# DomainGallery: Few-shot Domain-driven Image Generation by Attribute-centric Finetuning

**Yuxuan Duan**[1]    **Yan Hong**[2]    **Bo Zhang**[1]    **Jun Lan**[2]    **Huijia Zhu**[2]    **Weiqiang Wang**[2]
**Jianfu Zhang**[1*]    **Li Niu**[1*]    **Liqing Zhang**[1*]
[1]Shanghai Jiao Tong University    [2]Ant Group

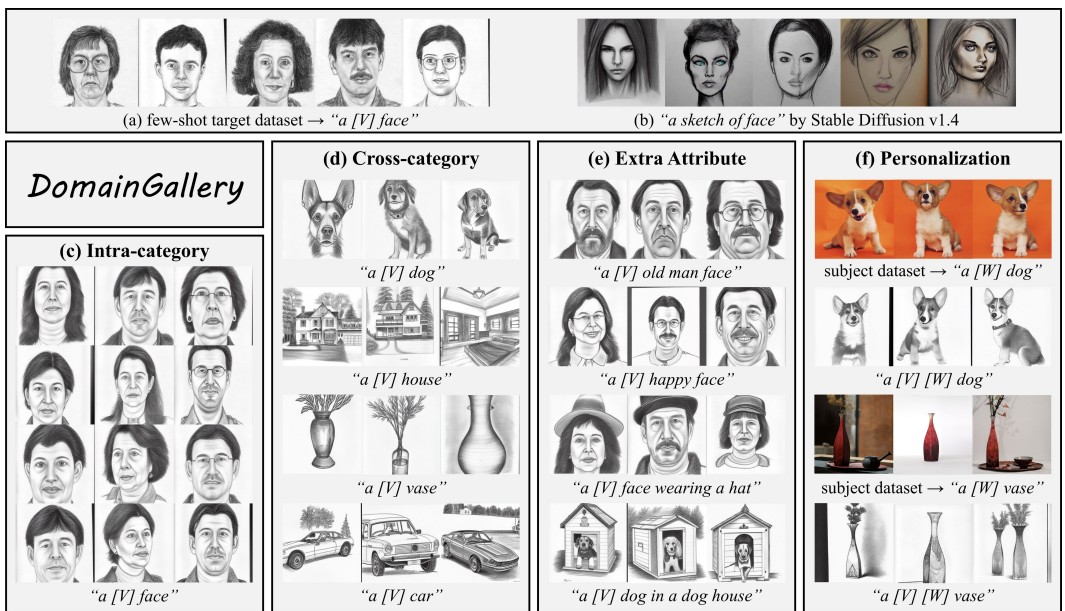

Figure 1: Given a few-shot target dataset of a specific domain such as sketches painted by an artist (a), it is usually difficult to directly generate images of this domain using pretrained text-to-image models (b). By using **DomainGallery** we propose in this work, we can achieve domain-driven generation in intra-category (c); cross-category (d); extra attribute (e); and personalization (f) scenarios.

## Abstract

The recent progress in text-to-image models pretrained on large-scale datasets has enabled us to generate various images as long as we provide a text prompt describing what we want. Nevertheless, the availability of these models is still limited when we expect to generate images that fall into a specific domain either hard to describe or just unseen to the models. In this work, we propose DomainGallery, a few-shot domain-driven image generation method which aims at finetuning pretrained Stable Diffusion on few-shot target datasets in an attribute-centric manner. Specifically, DomainGallery features prior attribute erasure, attribute disentanglement, regularization and enhancement. These techniques are tailored to few-shot domain-driven generation in order to solve key issues that previous works have failed to settle. Extensive experiments are given to validate the superior performance of DomainGallery on a variety of domain-driven generation scenarios.

---

*Corresponding authors.

38th Conference on Neural Information Processing Systems (NeurIPS 2024).

# 1  Introduction

Walking down the street, you see an artist painting portrait sketches for people. You are fascinated by a couple of masterpieces set by the side, showing his/her unique painting style which you find it difficult to describe by words. Deeply interested, you are in the mood for seeing more sketches, and it would be perfect to see him/her painting other things like dogs, especially your favorite ones at home.

As a fundamental topic in computer vision, image generation has been attracting enormous research efforts. However, through the years from VAEs [23], GANs [13] to diffusion models [16], generative models are becoming more and more data-hungry in order to properly model the distribution of images, as the most recent Stable Diffusions [35] have been trained on billions of text-image pairs [39]. Thus unfortunately, it is usually infeasible to directly train a generative model given only few-shot (around ten, or fewer) images of a specific target domain.

To tackle such challenging scenarios, one paradigm of solutions is *model transfer*, which first trains a model on a relevant source domain and then transfers it to the target domain by finetuning on the few-shot target dataset. Nevertheless, as Zhao et al. [56] have pointed out, the performance of model transfer methods will be significantly influenced by the relevance between source/target domains. Therefore, the applicability of these methods will be limited if we either fail to find a proper source dataset, or just do not have enough resources to train a generative model from scratch.

With the recent progress in pretrained text-to-image (T2I) models [28, 31–33, 35, 38, 52], it seems that anything can be generated simply by putting a text prompt into an off-the-shelf pretrained T2I model. However, T2I models are still far from once for all solutions to image generation. Sometimes it is difficult or even impossible to precisely describe certain styles (*e.g.* sketches by an artist) and contents (*e.g.* new concepts or personalized subjects), or what we want is simply unseen (thus unknown) to the model. Fortunately, T2I models can serve as *universal* source models to be finetuned on specific target datasets. Recent works finetuning T2I models have mostly focused on either finetuning with relatively abundant images (tens, hundreds, or more) [11], or few-shot subject-driven generation whose datasets consist of a single person or object [12, 37]. On the contrary, few-shot domain-driven generation analogous to the conventional model transfer has rarely been explored.

In this work, we analyze and perform few-shot domain-driven image generation from the view of *attributes*, as a domain is defined by common attributes shared among images (see Sec. 3). We seek to master four generation cases as illustrated in Fig. 1: **Intra-category:** The generated images contain both the domain attributes and the categorical attribute of the given target dataset, as in conventional model transfer; **Cross-category:** While containing non-categorical domain attributes, images of other categories can be generated through text control, as a feature of T2I models; **Extra attribute:** Either intra- or cross-category, we can attach additional attributes to the images; **Personalization:** We hope to combine domain-driven and subject-driven generation for better personalization. In order to achieve these goals, we propose **DomainGallery**, adopting DreamBooth-like [37] finetuning paradigm where the non-categorical domain attributes are learned and bound to an identifier word, so that the generation can be done via a normal T2I pipeline. DomainGallery features four attribute-centric finetuning techniques which respectively settle four challenges:

**(1) Prior attribute erasure:** The prior attributes of the identifier word may possibly show up even if we have bound new domain attributes to it. Therefore, we pre-erase these prior attributes to avoid unexpected elements in images.

**(2) Attribute disentanglement:** The domain/categorical attributes corresponding to the identifier/category word may be leaked into each other, causing missing domain attributes and/or unexpected categorical attributes when we change the category word in cross-category generation. Therefore, we explicitly encourage domain-category disentanglement to prevent such leakage.

**(3) Attribute regularization:** The model is prone to overfitting when finetuned on few-shot datasets. Therefore, we regularize the finetuning process (with a strategy to construct paired source/target latent codes and a regularization loss) to reduce overfitting caused by excessive presence of domain attributes and possible biases of dataset distributions.

**(4) Attribute enhancement:** Sometimes the strengths of the domain attributes learned on a specific dataset category are insufficient for cross-category generation. Therefore, we adjust the intensity of the domain attributes when generating cross-category images for better fidelity.

These techniques spreading over pre-finetuning (1)(2), finetuning (2)(3) and inference (4), are tailored to few-shot domain-driven generation, aiming at solving key issues that previous works have failed to settle. Later in Sec. 5, we conduct thorough experiments on several few-shot datasets. These experiments manifest the superior and satisfying performance of DomainGallery on all of the four generation scenarios, which can serve as a state-of-the-art method of few-shot domain-driven image generation.

## 2 Related Work

**Model Transfer**    Model Transfer (of conventional noise-to-image models instead of T2I ones) is a mainstream paradigm of solutions to few-shot image generation. Methods following this paradigm transfer models trained on related source datasets to target domains by finetuning on few-shot target datasets. Model transfer has been thoroughly explored using GANs [10, 18, 25–27, 29, 30, 34, 43, 46, 47, 50, 51, 55–58, 61], with a few base on diffusion models [20, 59]. Since T2I models came to light, people have been freed from choosing proper source datasets/models, and attention has been turned to finetuning T2I models as generic source models.

**Subject-driven Image Generation**    As one of the most frequently explored finetuning scenarios, subject-driven generation has attracted much research effort [2, 5, 7, 17, 24, 44, 48, 54] since the pioneering works Textual Inversion [12] and DreamBooth [37]. Actually, subject-driven generation can be categorized as a special case of domain-driven generation, where the domain is defined by a particular person or object. To preserve the subject identity, fidelity is highly preferred to diversity, as diversity is scarcely evaluated quantitatively by these works. On the contrary, in general domain-driven generation the domains are usually not confined to a specific subject. Therefore diversity is as important as fidelity, and we will evaluate both just as model transfer works do.

**Few-shot Domain-driven Image Generation**    We follow previous works to name our goal as few-shot domain-driven image generation. Analogous to *subject-driven*, the term *domain-driven* implies finetuning from T2I models, which enables us to take advantage of the multi-modal capability of these models to achieve a variety of generation scenarios (see Fig. 1(c–f)). To the best of our knowledge, there is only one previous work focusing on this topic, namely DomainStudio [60]. It finetunes a Stable Diffusion model towards the target domain by learning an identifier similar to DreamBooth [37], yet equipped with additional losses to enhance diversity and high-frequency details. In Sec. 4, we will analyze some crucial issues in domain-driven generation that previous works have failed to settle, and accordingly propose attribute-centric solutions to these problems.

**Other Similar Tasks**    There are some other works focusing on resembling tasks. For instance, Everaert et al. [11] have focused on finetuning under limited data (tens to hundreds) with per-image text prompts. Such requirement of image quantity and prompts has limited its applicability. Another similar topic is T2I style transfer [4, 6, 11, 40], which usually extracts style information from a single style image and controls the content via text. A key issue shared by these works is how to clearly defining the boundary between style and content from a single image. Instead, domains can be naturally delimited as the common attributes shared among multiple images, which also enables us to learn a domain of certain contents, rather than styles.

## 3 Preliminary

**Domain**    Formally, a *domain* $\mathcal{D}$ can be defined as a sample space $\mathcal{X}$ and a data distribution $P_{\text{data}}$ on $\mathcal{X}$ [1]. However, this definition is excessively general as any group of arbitrary images can form a domain. In this work, we would like to provide a rather intuitive definition from the viewpoint of common attributes. We regard an image $X$ to be composed of a set of attributes $\{a_i\}_{i=1}^N$, where each $a_i$ can be either abstract like a certain style, or concrete like a specific category or certain content. Then, an image domain $\mathcal{D}$ can be defined as the common attributes shared by all the images of this domain: $a_{\mathcal{D}} = \bigcap_{X \in \mathcal{D}} X$. According to such definition, an image belongs to this domain if and only if it contains all the common attributes: $X \in \mathcal{D} \iff a_{\mathcal{D}} \subseteq X$. Take the few-shot sketches of faces in Fig. 1(a) as an example, $a_{\mathcal{D}}$ includes shared categorical attribute of human faces and the attributes of this specific painting style, while the content attributes indicating individuals are not shared. Therefore, any facial sketch of any person in such style belongs to this domain.

Since in real-world scenarios, images in few-shot datasets usually share a common category (*e.g.* face in Fig. 1(a)), it is natural that categorical attribute should be one of the domain attributes. However, to extend domain-driven generation to cross-category scenarios as in Fig. 1(d), in this work we exclude the categorical attribute from $a_{\mathcal{D}}$ so that the domain attributes refer to non-categorical attributes only. For instance, the domain in Fig. 1 will be referred to as sketches (of anything) in this certain style.

**Diffusion Model**  Diffusion model [8, 16, 42] is a recent genre of generative models. It aims at reversing a diffusion process by recurrently predicting the noises based on noisy data and denoising them accordingly till proper images are rendered. For practical usage in high-resolution and conditional cases, Latent Diffusion Model (LDM) [35] is often adopted which moves the diffusion process to latent spaces with pretrained VAEs [23]. LDM is commonly trained using a simplified objective as

$$L_{\mathrm{LDM}} = \mathbb{E}_{l,c,\epsilon \sim \mathcal{N}(0,I),t} \left[ \| \epsilon - \epsilon_\theta(l_t, t, \tau_\theta(c)) \|_2^2 \right], \tag{1}$$

where $l$, $c$, $\epsilon$ and $t$ are respectively latent codes, conditions, ground-truth noises and time steps. The module $\tau_\theta$ is the encoder of the condition and $\epsilon_\theta$ is the noise-predicting network which is usually a UNet [36]. As special instances of LDM, Stable Diffusion (SD) series are pretrained on large-scale text-image datasets such as LAION-5B [39]. They serve as state-of-the-art T2I models that are widely used as base models in many tasks, including our DomainGallery as well.

**DreamBooth**  As a pioneering work in subject-driven image generation, DreamBooth [37] binds the information of the subject to an identifier [V], which is a rarely used word such as *sks*, together with a corresponding category word [N], such as *dog*. Then images of the target subject can be generated by using prompts like *"a [V] [N]"*. For domain-driven image generation, we inherit such design to bind (non-categorical) domain attributes to [V], so that by changing category words or adding extra attributes via text, DomainGallery is capable of generating various images within the given domain.

**Low-Rank Adaptation**  Low-Rank Adaptation (LoRA) [19] is a popular finetuning method frequently used on SD models. Instead of finetuning the parameters $\mathbf{W} \in \mathbb{R}^{d_{\mathrm{in}} \times d_{\mathrm{out}}}$, LoRA finetunes rank decomposition matrices $\mathbf{A} \in \mathbb{R}^{d_{\mathrm{in}} \times r}$ and $\mathbf{B} \in \mathbb{R}^{r \times d_{\mathrm{out}}}$ as in $\hat{\mathbf{W}} = \mathbf{W} + \mathbf{A} \cdot \mathbf{B}$, where $r$ is very small and $\mathbf{W}$ is fixed. Finetuned LoRA parameters can be easily shared and used with base models due to much smaller sizes. DomainGallery adopts LoRA when finetuning SD on target datasets.

## 4  DomainGallery

In this section, we will give a detailed description of DomainGallery. As in Fig. 2, the full pipeline has three steps: prior attribute erasure in Sec. 4.1, finetuning in Sec. 4.2, and inference in Sec. 4.4.

### 4.1  Prior Attribute Erasure

Following DreamBooth, we link target domain attributes to an identifier [V]. Although we expect to select a rarely used word without obvious meaning, this word may have still been bound to certain prior attributes. For instance, the commonly used *sks* is actually the abbreviation of a rifle [49], thus images generated with [V] in prompts will contain military elements, like the helmet in Fig. 2(a). In subject-driven generation such prior attributes are not problems, since the text condition *"a [V] [N]"*, as a whole, will gradually overfit to the given subject dataset and override these prior attributes. Also, [V] will never be paired with another category (*e.g.* *"a [V] cat"* when the subject is a dog), while in domain-driven generation we expect [V] to be applicable to any category. According to the results in Sec. 5.2 and Appendix B.1, if not pre-erased, these prior attributes will appear in cross-category images, which verifies that these prior attributes are merely concealed rather than eliminated, and it is necessary to erase them before usage.

Since the prior attributes are bound to the identifier in a data-driven manner when T2I models are pretrained, it is difficult to theoretically specify which attributes have been linked to [V]. Therefore, we propose an empirical solution to prior attribute erasure. Based on a noisy source latent $l_{\mathrm{src}}$ that has been added noise $\epsilon$ in the forward process, DomainGallery predicts the added noises $\epsilon_{\mathrm{src}}$ and $\epsilon_{\mathrm{s} \to \mathrm{t}}$ using the same LoRA-equipped UNet respectively with source text condition $c_{\mathrm{src}} = $ *"a [N]"* and target condition $c_{\mathrm{tgt}} = $ *"a [V] [N]"*. Then, the prior attribute erasure loss is defined as

$$L_{\mathrm{erase}} = \mathrm{MSE}(\epsilon_{\mathrm{s} \to \mathrm{t}}, \mathrm{gs}(\epsilon_{\mathrm{src}})), \quad \begin{cases} \epsilon_{\mathrm{src}} = \epsilon_{\theta,\phi}(l_{\mathrm{src}}, c_{\mathrm{src}}) \\ \epsilon_{\mathrm{s} \to \mathrm{t}} = \epsilon_{\theta,\phi}(l_{\mathrm{src}}, c_{\mathrm{tgt}}) \end{cases}, \tag{2}$$

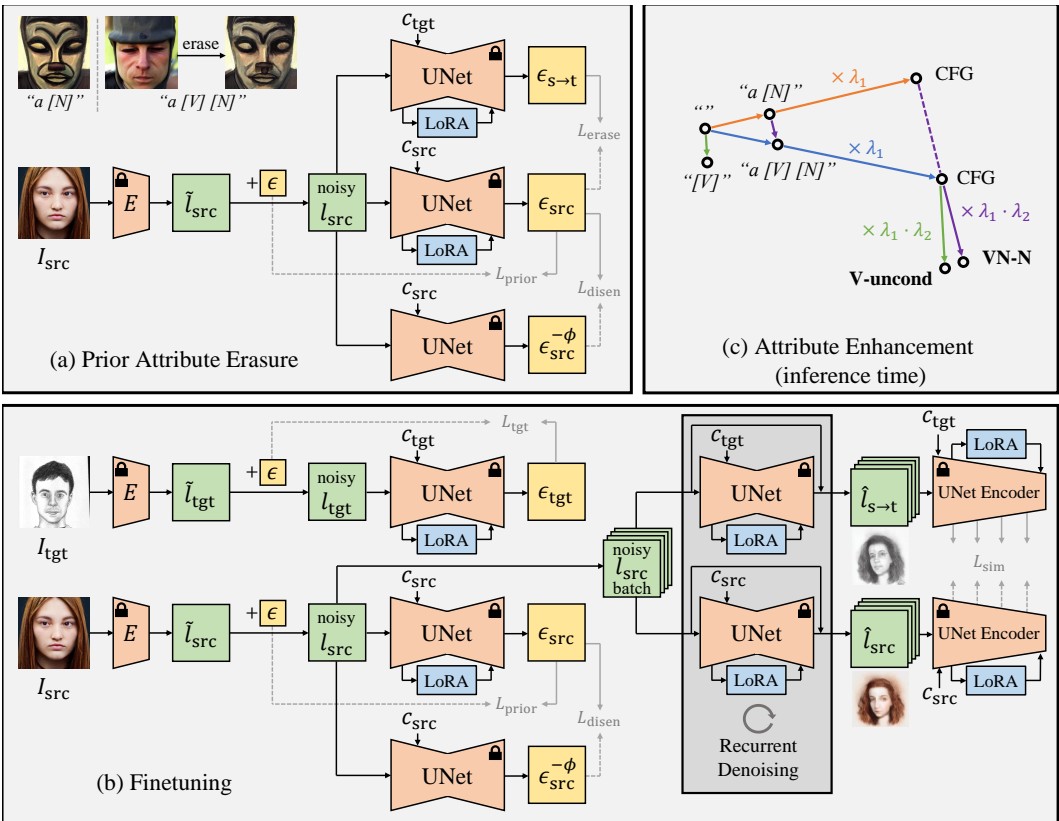

Figure 2: An overview of DomainGallery. **(a)** Before finetuning, we erase the prior attributes of the identifier [V] by matching the predicted noises when using source/target text conditions via $L_{\text{erase}}$. **(b)** During fintuning, besides training ordinarily on target datasets (top-left), we additionally impose domain-category attribute disentanglement loss $L_{\text{disen}}$ (bottom-left) and transfer-based similarity consistency loss $L_{\text{sim}}$ (right). **(c)** When generating cross-category images, we enhance the domain attributes referred by [V] in a CFG-like manner. Dashed arrows indicate gradient stopping.

where we omit time step $t$ and text encoder $\tau$ in the UNet $\epsilon_{\theta,\phi}$ for brevity, $\phi$ indicates LoRA parameters and $\text{gs}(\cdot)$ is the gradient stopping operation that stops the gradient from propagating through or updating the parameters inside. By imposing $L_{\text{erase}}$, we hope that the model predicts the same with or without [V], hence the prior attributes in [V] will be removed.

Besides $L_{\text{erase}}$, the prior preservation loss $L_{\text{prior}}$ of DreamBooth is also applied which trains on source images $I_{\text{src}}$ generated by the base model of SD itself as training a diffusion model ordinarily via Eq. (1). Also, disentanglement loss $L_{\text{disen}}$ is also included, which will be detailed in Sec. 4.2. After erasure, the learned LoRA parameters $\phi$ will be used to initialize LoRA in the finetuning period.

## 4.2 Finetuning

With prior attributes of [V] erased, DomainGallery then learns to bind the target domain attributes to [V]. In addition to a standard finetuning on target datasets by $L_{\text{tgt}}$ via Eq. (1), with prior preservation on pre-generated source images, we propose domain-category attribute disentanglement loss $L_{\text{disen}}$ and transfer-based similarity consistency loss $L_{\text{sim}}$, as depicted in Fig. 2(b).

**Domain-category Attribute Disentanglement** Since few-shot datasets usually share a common category (*e.g.* face in Fig. 1(a)), when finetuning on such datasets, the (non-categorical) domain attributes in [V] will always show up together with the categorical attribute in [N], both in target images $I_{\text{tgt}}$ and target prompts $c_{\text{tgt}}$. As a result, it is possible that certain domain attributes may leak into [N], and/or conversely the categorical attribute may leak into [V]. Although it is not a

problem either for subject-driven generation since [V] and [N] will always be paired when generating images, such entanglement between [V] and [N] will harm cross-category scenarios of domain-driven generation. As experimental results shown in Sec. 5.2 and Appendix B.1, if we replace [N] with another category, sometimes domain attributes are partially lost, or elements of the original category still appear.

To tackle this issue, we try to enhance the disentanglement between [V] and [N], so that all the domain attributes will only be learned into [V] without leaking into [N], and categorical attributes in [N] will not be lost. In other words, attributes of [N] after finetuning should not be different from those before. As we use LoRA, the base model before finetuning is ready to use by simply disenabling LoRA parameters $\phi$ temporarily since the UNet is fixed. Based on noisy source latent $l_{\text{src}}$ and source text condition $c_{\text{src}}$, the domain-category attribute disentanglement loss can be formulated as

$$L_{\text{disen}} = \text{MSE}(\epsilon_{\text{src}}, \text{gs}(\epsilon_{\text{src}}^{-\phi})), \quad \begin{cases} \epsilon_{\text{src}} = \epsilon_{\theta,\phi}(l_{\text{src}}, c_{\text{src}}) \\ \epsilon_{\text{src}}^{-\phi} = \epsilon_{\theta}(l_{\text{src}}, c_{\text{src}}) \end{cases}, \tag{3}$$

where $\epsilon_{\theta}$ without $\phi$ is the base UNet whose LoRA parameters are detached.

**Attribute Regularization**   Adding regularization is a common practice of model transfer methods [30, 51, 57] to prevent overfitting, where features from **paired** source/target images generated from the same noise are usually required. However, according to the training objective of SD in Eq. (1), no fully denoised latent (*i.e.* at time step 0) will be generated, let alone paired source/target latents. DomainStudio [60] has proposed a regularization, which applies a similarity consistency loss [30] on batches of source/target images $\hat{I}_{\text{src}}/\hat{I}_{\text{tgt}}$ decoded from denoised latents $\hat{l}_{\text{src}}/\hat{l}_{\text{tgt}}$ after a single-step denoising from noisy latents $l_{\text{src}}/l_{\text{tgt}}$. However, there are four drawbacks in this design: **(1)** single-step denoising usually does not lead to meaningful latents/images unless the timestep is small; **(2)** decoding latents into images induces significant overhead of computation and storage; **(3)** computing cosine similarity between pixel-level images is less reasonable; **(4)** $\hat{I}_{\text{src}}/\hat{I}_{\text{tgt}}$ are unpaired as they derives from unpaired input source/target images $I_{\text{src}}/I_{\text{tgt}}$, which do not fit the similarity consistency loss requiring paired images/features. In our DomainGallery, we propose a strategy of constructing paired source/target latents, followed by a new regularization term named transfer-based similarity consistency loss, which overcomes the aforementioned drawbacks.

First we try to settle **(1)** and **(4)** by constructing denoised, meaningful and paired latent codes. As in the right part of Fig. 2(b), given a batch of $l_{\text{src}}$ at time step $t$, we conduct an $n$-step recurrent denoising following the accelerated denoising process of DDIM [42] and a linearly decreasing time step schedule from $t$ to 0. We intuitively set $n = 5$ to balance denoising quality and speed. We do recurrent denoising for twice, respectively with source/target text $c_{\text{src}}/c_{\text{tgt}}$, and obtain $\hat{l}_{\text{src}}/\hat{l}_{\text{s}\to\text{t}}$. If we decode them into images $\hat{I}_{\text{src}}/\hat{I}_{\text{s}\to\text{t}}$, we will find that $\hat{I}_{\text{s}\to\text{t}}$ simultaneously have partial target domain attributes after conditioned on $c_{\text{tgt}}$, and share certain similarity with $\hat{I}_{\text{src}}$ since they derive from the same $l_{\text{src}}$. Hence $\hat{l}_{\text{src}}/\hat{l}_{\text{s}\to\text{t}}$ are paired. Next, **without actually decoding them into images**, we reuse the encoder of UNet as a pretrained feature extractor to directly extract multi-layer features from $\hat{l}_{\text{src}}$ and $\hat{l}_{\text{s}\to\text{t}}$, and compute the similarity consistency loss as

$$L_{\text{sim}} = \frac{1}{N \cdot B} \sum_{k=1}^{N} \sum_{i=1}^{B} D_{\text{KL}}\left(p_{\text{s}\to\text{t}}^{i,k} \| \text{gs}(p_{\text{src}}^{i,k})\right),$$

$$p_{\text{s}\to\text{t}}^{i,k} = \text{Softmax}\left(\{\text{CosSim}(f^k(\hat{l}_{\text{s}\to\text{t}}^i, c_{\text{tgt}}), f^k(\hat{l}_{\text{s}\to\text{t}}^j, c_{\text{tgt}}))\}_{\forall j \neq i}\right), \tag{4}$$

$$p_{\text{src}}^{i,k} = \text{Softmax}\left(\{\text{CosSim}(f^k(\hat{l}_{\text{src}}^i, c_{\text{src}}), f^k(\hat{l}_{\text{src}}^j, c_{\text{src}}))\}_{\forall j \neq i}\right),$$

where $f^k$ represents features extracted by the UNet encoder at its $k$-th layer (of $N$ layers). From the viewpoint of the $i$-th latent (of $B$ latents in the batch), first its cosine similarities with other latents are computed, followed by a softmax operation transforming them into a probabilistic distribution $p^i$. Then, Kullback-Leibler Divergence will be computed between two distributions respectively from $\hat{l}_{\text{src}}^i$ and $\hat{l}_{\text{s}\to\text{t}}^i$, and will be averaged among all $i$ and layers $k$. In conclusion, $L_{\text{sim}}$ helps to prevent overfitting by matching the similarity distributions among a batch of paired $\hat{l}_{\text{src}}/\hat{l}_{\text{s}\to\text{t}}$. By operating on the features directly extracted from latents rather than on images, our $L_{\text{sim}}$ settles **(2)** and **(3)** as well.

### 4.3 Objective

The objective functions of prior attribute erasure and finetuning are respectively

$$L_{\text{erasure}} = L_{\text{prior}} + \lambda_{\text{disen}} \cdot L_{\text{disen}} + \lambda_{\text{erase}} \cdot L_{\text{erase}},$$
$$L_{\text{finetune}} = L_{\text{tgt}} + \lambda_{\text{prior}} \cdot L_{\text{prior}} + \lambda_{\text{disen}} \cdot L_{\text{disen}} + \lambda_{\text{sim}} \cdot L_{\text{sim}}, \tag{5}$$

where $\lambda_{\text{prior}} = 1.0$, $\lambda_{\text{disen}} = 10.0$, $\lambda_{\text{erase}} = 10.0$, $\lambda_{\text{sim}} = 1.0$ generally renders good results. Note that as we utilize LoRA in DomainGallery, only the additional parameters $\phi$ of LoRA will be updated.

### 4.4 Inference

In preliminary cross-category experiments, the domain attributes are not sufficiently manifested sometimes. A possible reason is that $L_{\text{sim}}$ has limited the strengths of these attributes to the minimal, just enough to transfer images of the original category, while for cross-category scenarios these attributes may need enhancing. As shown in Fig. 2(c), we propose an inference-time attribute enhancement based on classifier-free guidance (CFG)[15]. Specifically, after applying CFG with default weight $\lambda_1 = 7.5$, we additionally increase the strength of [V] by either of

$$\textbf{VN-N: } \epsilon = \epsilon(\text{``''}) + \lambda_1(\epsilon(\text{``}a\,[V]\,[N]\text{''}) - \epsilon(\text{``''})) + \lambda_1 \cdot \lambda_2(\epsilon(\text{``}a\,[V]\,[N]\text{''}) - \epsilon(\text{``}a\,[N]\text{''}));$$
$$\textbf{V-uncond: } \epsilon = \epsilon(\text{``''}) + \lambda_1(\epsilon(\text{``}a\,[V]\,[N]\text{''}) - \epsilon(\text{``''})) + \lambda_1 \cdot \lambda_2(\epsilon(\text{``}[V]\text{''}) - \epsilon(\text{``''})). \tag{6}$$

Between the two enhancing modes above, we empirically find that V-uncond generally outperforms its counterpart (see Appendix B.1). We will by default apply V-uncond with $\lambda_2 = 1.0$ during cross-category generation.

### 4.5 Personalization

For personalization scenarios, we straightforwardly combine our DomainGallery with DreamBooth in a single stage. Specifically, during the finetuning process in Sec. 4.2, the model is additionally finetuned on target subject images and source images of subject category via Eq. (1). In such cases, the objective of finetuning in Eq. (5) will be rewritten as

$$L_{\text{person}} = L_{\text{finetune}} + \lambda_{\text{subject}} \cdot (L_{\text{tgt}}^{\text{subject}} + \lambda_{\text{prior}} \cdot L_{\text{prior}}^{\text{subject}}), \tag{7}$$

where $\lambda_{\text{subject}}$ is empirically set to 1.0. While we suppose that there may be a more delicate way to equip DomainGallery with subject-driven methods, we would like to leave it for future works.

## 5 Experiment

### 5.1 Experimental Setting

**Baseline**  Our baseline list includes DreamBooth [37], as the basis of our method; a LoRA [19] version of DreamBooth, since we utilize LoRA in DomainGallery; and finally DomainStudio [60], as the only previous work in few-shot domain-driven image generation.

**Dataset**  We test our method on five widely used 10-shot datasets, including CUFS sketches [45] ([N]: *face*), FFHQ sunglasses [21] ([N]: *face*), Van Gogh houses [30] ([N]: *house*), watercolor dogs [41] ([N]: *dog*) and wrecked cars [30] ([N]: *car*). Note that though sunglasses and wrecked cars may also be generated by directly mentioning their content attributes in text prompts, we still try on these datasets to prove that DomainGallery can also learn content attributes. Experiments are conducted on resolution $512 \times 512$ except for DomainStudio which is only capable of $256 \times 256$ even on a 40GB VRAM GPU.

**Metric**  We provide quantitative results for intra-category generation since we have dataset images as ground truth. For datasets with full sets (CUFS sketches and FFHQ sunglasses), we compute FID [14] between 1,000 samples with the full sets. For the others, we replace FID with KID [3] ($\times 10^3$) which better fits few-shot scenarios [9, 10, 22, 56, 58]. Intra-clustered LPIPS [30, 53] of 1,000 samples with the few-shot training sets is also reported as a standalone diversity metric.

**Detail**  For other details of the experiments and DomainGallery, please refer to Appendix A.

Table 1: Quantitative results of the intra-category scenarios on CUFS **sketches**, FFHQ **sunglasses**, Van Gogh **houses**, watercolor **dogs** and wrecked **cars**. The underlined results of DreamBooth have severe overfitting issues hence achieve good KID scores, see qualitative results in Fig. 10.

| Method | sketches | | sunglasses | | houses | | dogs | | cars | |
|---|---|---|---|---|---|---|---|---|---|---|
| | FID↓ | I-LPIPS↑ | FID↓ | I-LPIPS↑ | KID↓ | I-LPIPS↑ | KID↓ | I-LPIPS↑ | KID↓ | I-LPIPS↑ |
| DreamBooth [37] | 70.41 | 0.4609 | 44.90 | 0.6451 | 48.43 | 0.6882 | **32.60** | 0.4005 | **8.81** | 0.5661 |
| DreamBooth+LoRA | 52.80 | 0.4636 | **41.22** | 0.6452 | 44.51 | 0.6744 | 68.80 | 0.4992 | 26.68 | 0.6063 |
| DomainStudio [60] | 51.73 | 0.4184 | 66.66 | 0.6089 | 41.06 | 0.6367 | 71.33 | 0.4059 | 32.31 | 0.5577 |
| **DomainGallery** | **44.86** | **0.5060** | 43.10 | **0.6924** | **32.20** | **0.7255** | 61.95 | **0.5216** | 23.63 | **0.6336** |

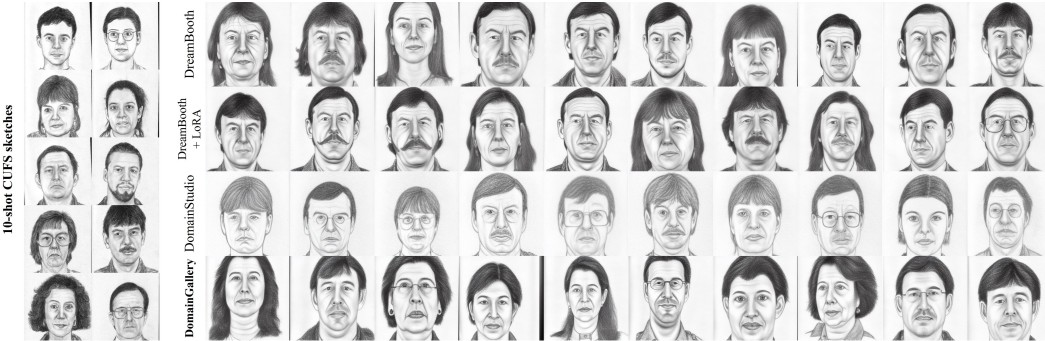

Figure 3: The 10-shot CUFS sketches dataset (left) and the intra-category samples generated by the baselines and DomainGallery with prompt *"a [V] face"* (right).

## 5.2 Experimental Result

**Intra-category** As the most basic scenario, we generate target images of the original categories. According to Tab. 1, DomainGallery generally outperforms the baselines w.r.t. both fidelity and diversity. These scores also match the qualitative results on CUFS sketches in Fig. 3, where Domain-Gallery can precisely capture the painting style of the target domain. Also, due to the effectiveness of our transfer-based similarity consistency loss $L_{\text{sim}}$, the diversity of DomainGallery surpasses the baselines by large margins, while achieving competitive or even better fidelity. Refer to Fig. 10 in Appendix B.2 for qualitative results on the other datasets.

**Cross-category** We illustrate qualitative results of cross-category generation on Van Gogh houses and watercolor dogs in Fig. 4. Since no previous method has pre-erased the prior attributes of [V] (*sks*) before usage, prior attributes of military elements can be observed in the samples generated by all the baselines. Besides, as none of the baselines explicitly imposes disentanglement between [V] and [N], attribute leakage can be observed on both datasets. Some images of DomainStudio still contain houses even if we change [N], manifesting leaked categorical attribute in [V]. On the other hand, many cross-category images of the baselines do not properly depict target domain attributes while their intra-category images do in Appendix B.2. Such phenomenon verifies that domain attributes have been partially leaked into [N] and will disappear if we change it. By adopting prior attribute erasure and enhancing domain-category attribute disentanglement, our DomainGallery avoids these issues and performs well. Samples on the other datasets are shown in Fig. 11 in Appendix B.2.

**Extra Attribute** In Fig. 5 and Fig. 1(e), we show some images generated by DomainGallery on CUFS sketches with extra attributes added to either intra- or cross-category scenarios. We may infer from these results that though we only provide simple prompt (*i.e. "a [V] [N]"*) rather than detailed description for each image of the target dataset, training DomainGallery does not destruct the original text-image structures of SD. The images are still under full control through text prompts, including facial expressions, additional contents (*e.g.* accessories), sub-category (*e.g.* a breed of animals), background, and specific instances (*e.g.* celebrities or brands).

Besides, the bottom row of Fig. 5 illustrates some samples where the extra attributes (*blue*) provided in the text prompt are in conflict with the domain attributes (*colorless*). In such case, DomainGallery is capable of generating images with partially fused attributes. While the images are generally in

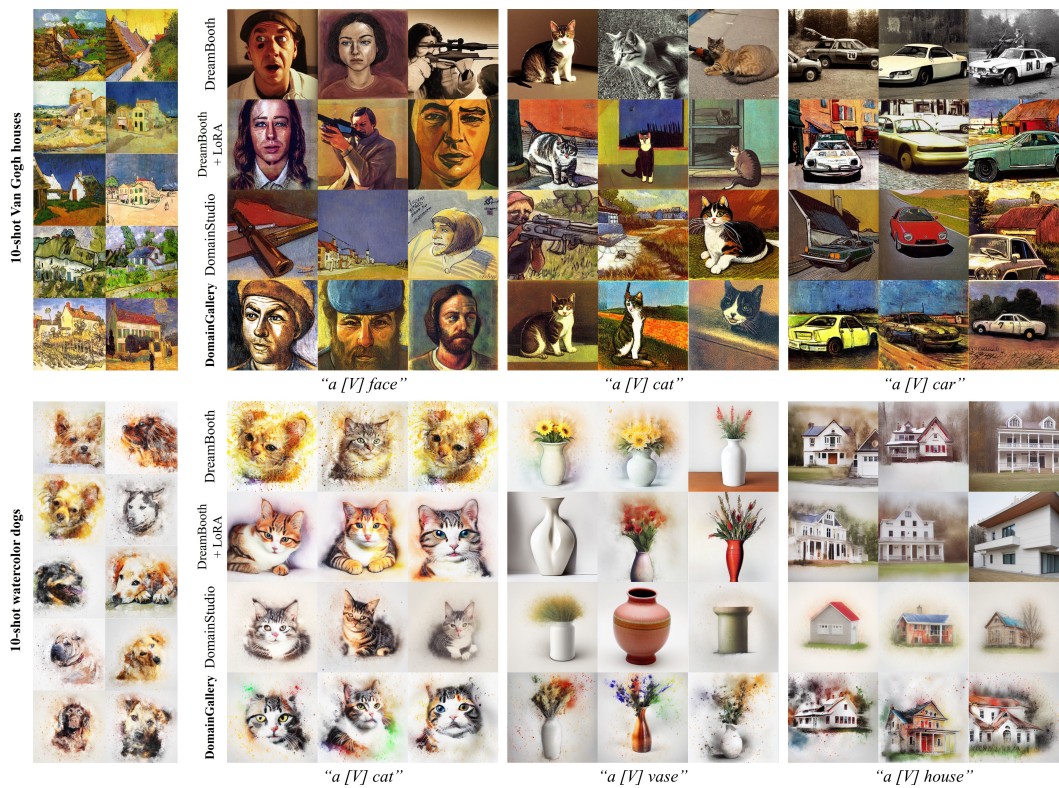

Figure 4: The 10-shot datasets (left) and the cross-category samples generated by the baselines and DomainGallery (right), on Van Gogh houses (top) and watercolor dogs (bottom).

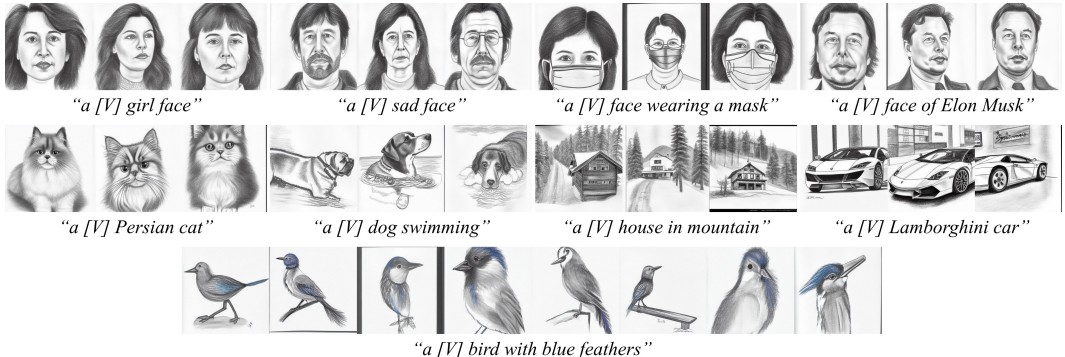

Figure 5: Intra-category (top row) and cross-category (middle row) samples with extra attributes given by texts generated by DomainGallery, on CUFS sketches. The bottom row additionally show the case where the text contains conflicting attributes.

grayscale, some blue feathers still appear. These results verify the generalization ability of our method and suggest that it may be open to other generation scenarios such as local editing and style blending.

**Personalization**   In the last scenario, DomainGallery is combined with DreamBooth to learn a target domain and a target subject simultaneously, as described in Sec. 4.5. Results in Fig. 6 manifest that such combination is feasible for both intra-category (the target dataset and subject share the same category, *e.g.* watercolor dogs and the subject of specific dog) and cross-category (otherwise) pairs of datasets. Together with the results of the previous scenario with extra attributes, the satisfying performance of DomainGallery shows its potentials to be applied to parallel or downstream tasks.

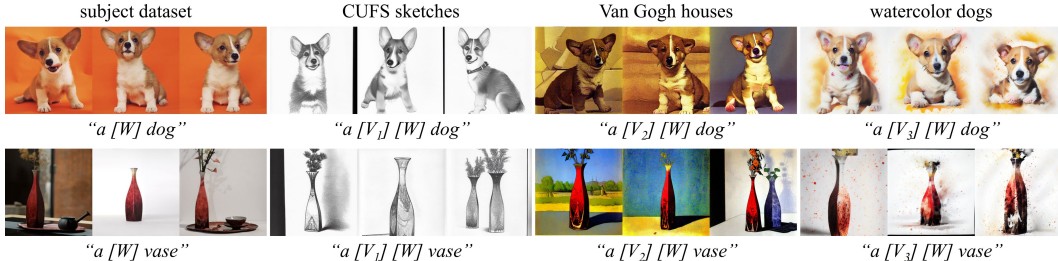

Figure 6: Few-shot subject datasets (left, partially shown) and the personalized samples generated by DomainGallery on CUFS sketches, Van Gogh houses and watercolor dogs.

**Ablation Study**  To prove that the proposed attribute-centric techniques can indeed effectively improve the performance of DomainGallery in various generation scenarios, we conduct extensive ablation studies focusing on these techniques and leave them in Appendix B.1 due to page limit.

# 6   Conclusion

In this work, we focus on few-shot domain-driven image generation by analyzing several key issues that previous works have failed to settle, and accordingly proposing a new method named Domain-Gallery. DomainGallery features four attribute-centric finetuning techniques that aim at solving these issues, namely prior attribute erasure, attribute disentanglement, attribute regularization and attribute enhancement. With these designs tailored to domain-driven generation, our DomainGallery achieves convincing performance on both intra-category and cross-category generation scenarios, while supporting extra attributes added by text prompts. Additionally, DomainGallery can be aggregated with subject-driven generation as well, which further extends its applicability. In Appendix C, we will discuss possible limitations and potential future works of DomainGallery.

# Acknowledgements

The work was supported by the National Science Foundation of China (62076162, 62471287, 62302295), and the Shanghai Municipal Science and Technology Major Project, China (2021SHZDZX0102). This work was also supported by Ant Group.

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

# Appendix

## A  Implementation Detail

**Model**   Our DomainGallery takes Stable Diffusion (SD) [35] as its base model. For the sake of fairness, DomainGallery and all the baselines share a common base model of SD v1.4,[2] though DomainGallery is probably applicable to newer versions since the attribute-centric techniques proposed in this work are not based on specific structures of the current version.

During the periods of prior attribute erasure and finetuning, we apply LoRA [19] of PEFT[3] to the UNet of SD, with rank $r = 4$, and on parameters of *to_k*, *to_q*, *to_v*, *to_out.0*, *add_k_proj* and *add_v_proj* by default. We do not finetune the text encoder $\tau$ or apply LoRA to it.

**Training**   For prior attribute erasure, we train the model for 500 steps, with batch size 4 and learning rate $1 \times 10^{-4}$. While for finetuning, we initialize LoRA with the parameters $\phi$ where prior attributes of the identifier [V] are erased, and train the model for 1,000 steps, with batch size 4 and learning rate $5 \times 10^{-5}$. During both periods, gradient checkpointing and 8bit Adam[4] are also applied to save VRAM. All the experiments running DomainGallery in this work are done on a single NVIDIA RTX 4090 GPU with 24GB VRAM.

For finetuning, we additionally apply offset noise[5] on CUFS sketches and watercolor dogs, as their images are obviously lighter than average.

**Inference**   When generating images during inference period, we apply DDIM [42] scheduler with 50 steps and scale of CFG [15] $\lambda_1 = 7.5$. When generating cross-category images, attribute enhancement is also applied as Sec. 4.4. Note that delicately selecting a scheduler and its parameters may render better images, however it is beyond the scope of this work.

## B  Additional Experiment

### B.1  Ablation Study

In this section, we provide ablation studies regarding the four attribute-centric techniques (prior attribute erasure, attribute disentanglement, attribute regularization, and attribute enhancement) proposed in this work, to prove that these techniques are indeed effective to few-shot domain-driven generation.

**Prior Attribute Erasure**   In Fig. 7(top) we ablate the process of prior attribute erasure described in Sec. 4.1, and generate some cross-category images after finetuning. Compared with the full DomainGallery in Fig. 7(bottom), military elements can be commonly observed, as these prior attributes of the identifier *sks* have been kept. Hence, pre-erasing the prior attributes of [V] is necessary before finetuning.

**Attribute Disentanglement**   We remove the domain-category attribute disentanglement loss $L_{\text{disen}}$ in Sec. 4.2 during finetuning and generate some cross-category images in Fig. 7(middle). Without $L_{\text{disen}}$ enhancing disentanglement, the domain attributes are partially lost when we change the category word [N], as some images do not present proper styles. On the other hand, the original category has been leaked into [V], as human faces (or patterns like human faces) appear in the images sometimes, though we have changed [N]. These phenomena necessitate the enhancement of disentanglement in DomainGallery.

**Attribute Regularization**   In Fig. 8, we illustrate the trend of FID and I-LPIPS scores on CUFS sketches when we use different weights $\lambda_{\text{sim}}$ between 0.0 and 2.0 for the transfer-based similarity consistency loss $L_{\text{sim}}$. As the results shown, the diversity of the generated images is indeed improving

---

[2] https://huggingface.co/CompVis/stable-diffusion-v1-4
[3] https://github.com/huggingface/peft
[4] https://huggingface.co/docs/transformers/main/en/perf_train_gpu_one
[5] https://www.crosslabs.org/blog/diffusion-with-offset-noise

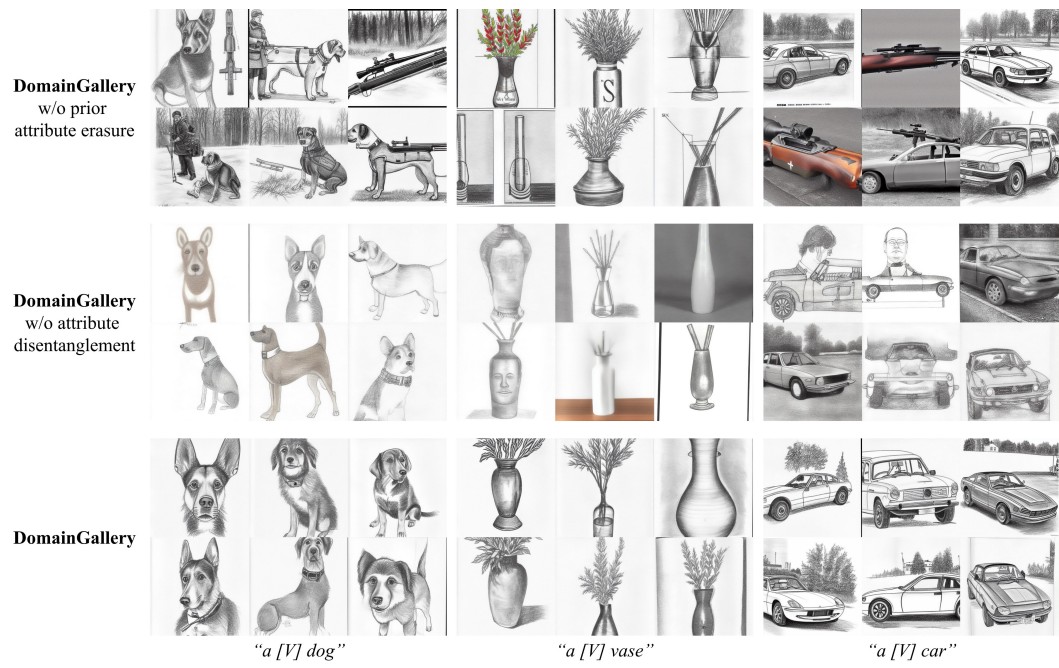

DomainGallery
w/o prior
attribute erasure

DomainGallery
w/o attribute
disentanglement

DomainGallery

*"a [V] dog"*       *"a [V] vase"*       *"a [V] car"*

Figure 7: The cross-category samples generated by DomainGallery without prior attribute erasure (top), DomainGallery without attribute disentanglement (middle), and the full DomainGallery (bottom) on CUFS sketches.

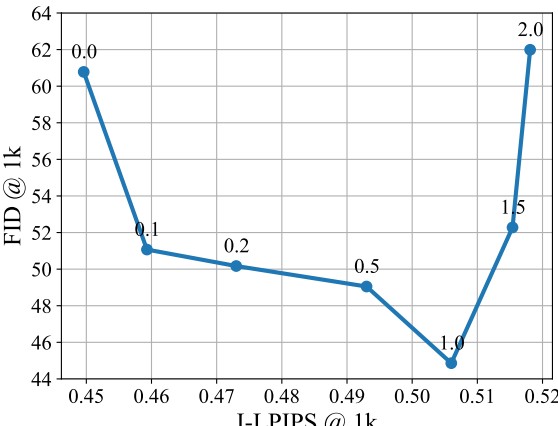

Figure 8: FID and I-LPIPS scores achieved by DomainGallery with different $\lambda_{\text{sim}}$ (annotated above the corresponding data points) ranging from $0.0$ to $2.0$, on CUFS sketches.

as we increase $\lambda_{\text{sim}}$. Besides, since $L_{\text{sim}}$ can prevent the model from learning unnecessary attributes induced by the bias of the few-shot datasets, it also enhances the fidelity of the generated images. However when the weight exceeds $1.0$, the regularization inhibits the model from learning necessary domain attributes as the fidelity begins to deteriorate.

**Attribute Enhancement** As the last part of the ablation study we investigate the effects of attribute enhancement in Sec. 4.4 during inference time. First we try to apply our attribute enhancement to the three baselines (DreamBooth [37], DreamBooth + LoRA, and DomainStudio [60]). As the top three rows of Fig. 9 show, enhancing [V] alone cannot improve the fidelity of the images, unless we

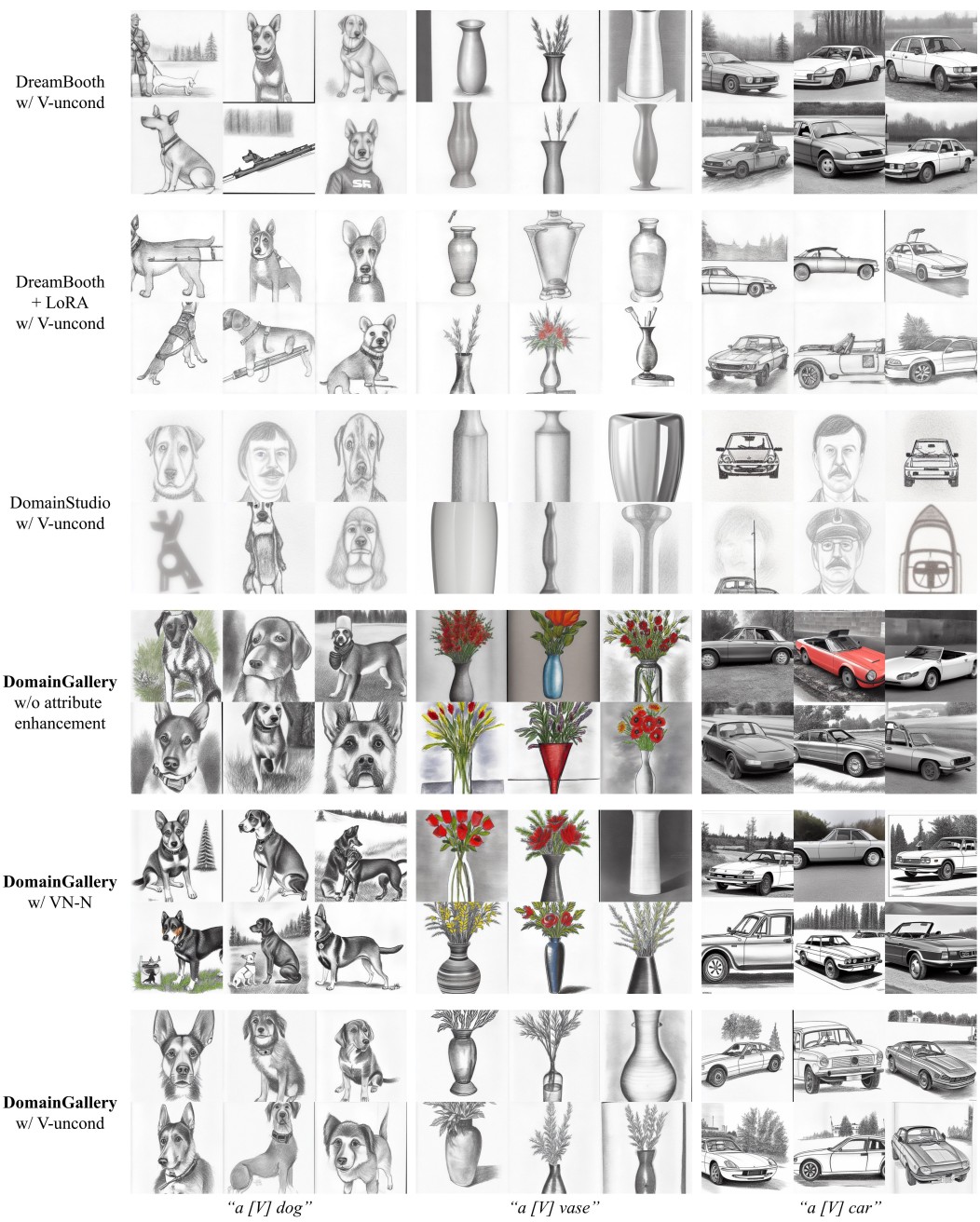

DreamBooth w/ V-uncond

DreamBooth + LoRA w/ V-uncond

DomainStudio w/ V-uncond

**DomainGallery** w/o attribute enhancement

**DomainGallery** w/ VN-N

**DomainGallery** w/ V-uncond

*"a [V] dog"*          *"a [V] vase"*          *"a [V] car"*

Figure 9: The cross-category samples generated by the three baselines with attribute enhancement (top three rows), by DomainGallery without attribute enhancement (fourth row), and by DomainGallery with attribute enhancement of either VN-N or V-uncond mode (last two rows) on CUFS sketches.

properly learn the target domain attributes into [V] in the first place, as our DomainGallery does (see the bottom row of Fig. 9).

Besides, as we have proposed two modes of attribute enhancement in Sec. 4.4, we would like to make a comparison between them. In the last two rows of Fig. 9 we illustrate samples generated following either mode. Although both modes can enhance the domain attributes to certain extents compared to DomainGallery without attribute enhancement in the fourth row of Fig. 9, the mode of V-uncond generally performs better than its counterpart. Therefore, we utilize V-uncond in DomainGallery by default when generating cross-category images.

**B.2   Additional Result**

In this section, we present additional qualitative results that are not illustrated in the main paper due to page limit.

**Intra-category**   Besides the intra-category images on CUFS sketches shown in Fig. 3, we depict those on the other datasets in Fig. 10. Generally, our DomainGallery surpasses the baselines on all the datasets. It is also worth mentioning that in few-shot domain-driven generation, the domains are not limited to certain styles (as in CUFS sketches, Van Gogh houses and watercolor dogs). Instead, our method is also applicable to domains of certain contents (FFHQ sunglasses and wrecked cars).

**Cross-category**   Cross-category images of the other datasets not shown in Fig. 4 of the main paper are in Fig. 11. Similar to the results in the main paper, elements of the prior attributes of [V], and attribute leakage between [V] and [N] can also be observed in these images, which necessitates prior attribute erasure and attribute disentanglement proposed in DomainGallery.

## C   Limitation and Future Work

In this work, we propose DomainGallery, a new method for few-shot domain-driven image generation. Although the experiments in Sec. 5 and Appendix B have validated the capability of DomainGallery, there are still some limitations w.r.t. the availability of our method which indicate directions for future works, as discussed below.

- Our method may not be able to handle the cases where the datasets consist images of different categories (*e.g.* a set of paintings of various objects by a certain artist), since DomainGallery follows DreamBooth that finetunes on a single category word [N]. However, with minor modification to the DreamBooth-like finetuning pipeline, DomainGallery may be capable of such cases by using per-image category words.

- Although we assume that domains should be defined based on obvious common attributes, sometimes there are composite domains that include several sub-domains (*e.g.* portraits painted by several artists of the Renaissance). In such cases the common attributes among all the images may be subtle and hard to tell. Therefore, for few-shot domain-driven methods (not limited to DomainGallery), domains with clear common attributes are preferred.

- Currently the performance of DomainGallery on domains of contents (*e.g.* FFHQ sunglasses) is still in need of further improvement, as we admit that the cross-category images on FFHQ sunglasses shown in Fig. 11 have undergone some cherry-picking. We suppose that it is much more difficult to finetune models on few-shot datasets of certain local contents than global styles (*e.g.* CUFS sketches), since semantic relations between the contents and the backgrounds (*e.g.* where to put sunglasses on faces, or even on faces of animals) can only be well learned through rather adequate data. Therefore, how to make few-shot domain-driven methods master on content domains is another direction for future works.

## D   Broader Impact

As a new method for few-shot domain-driven image generation, DomainGallery can be used either in creative AI applications, or generating image data as a non-traditional data augmentation for various downstream tasks. Furthermore, as image generation is a fundamental task in computer vision, the idea of DomainGallery may also be applied to researches in other topics.

However, similar to all the methods of image generation (including but not limited to few-shot domain-driven image generation), our method may induce possible societal harms, including fake image generation for misuse and copyright violation, depending on the specific applications. Therefore, we hereby request proper usage of DomainGallery.

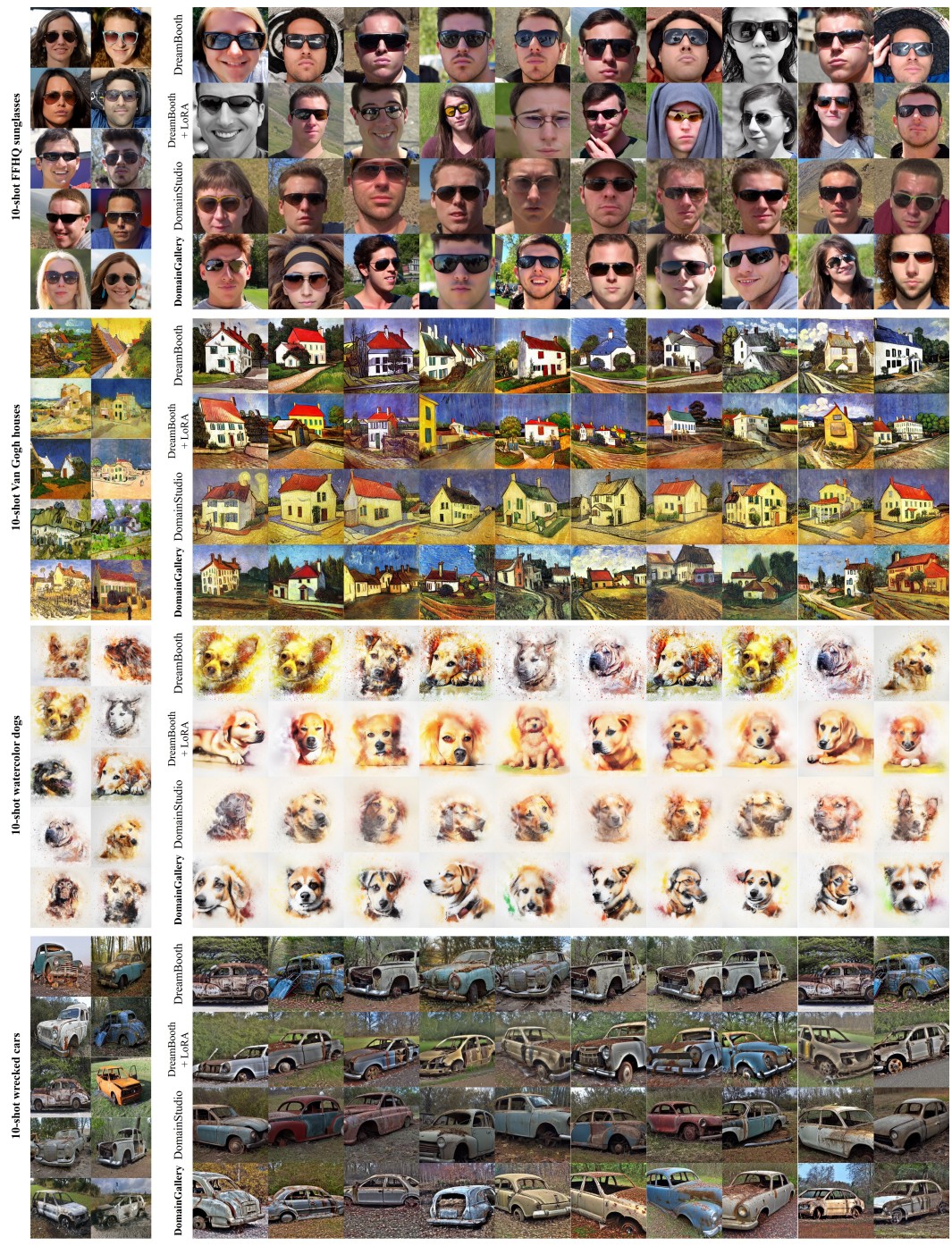

Figure 10: The 10-shot datasets (left) and the intra-category samples generated by the baselines and DomainGallery (right), respectively on FFHQ sunglasses (*"a [V] face"*), Van Gogh houses (*"a [V] house"*), watercolor dogs (*"a [V] dog"*) and wrecked cars (*"a [V] car"*) (from top to bottom).

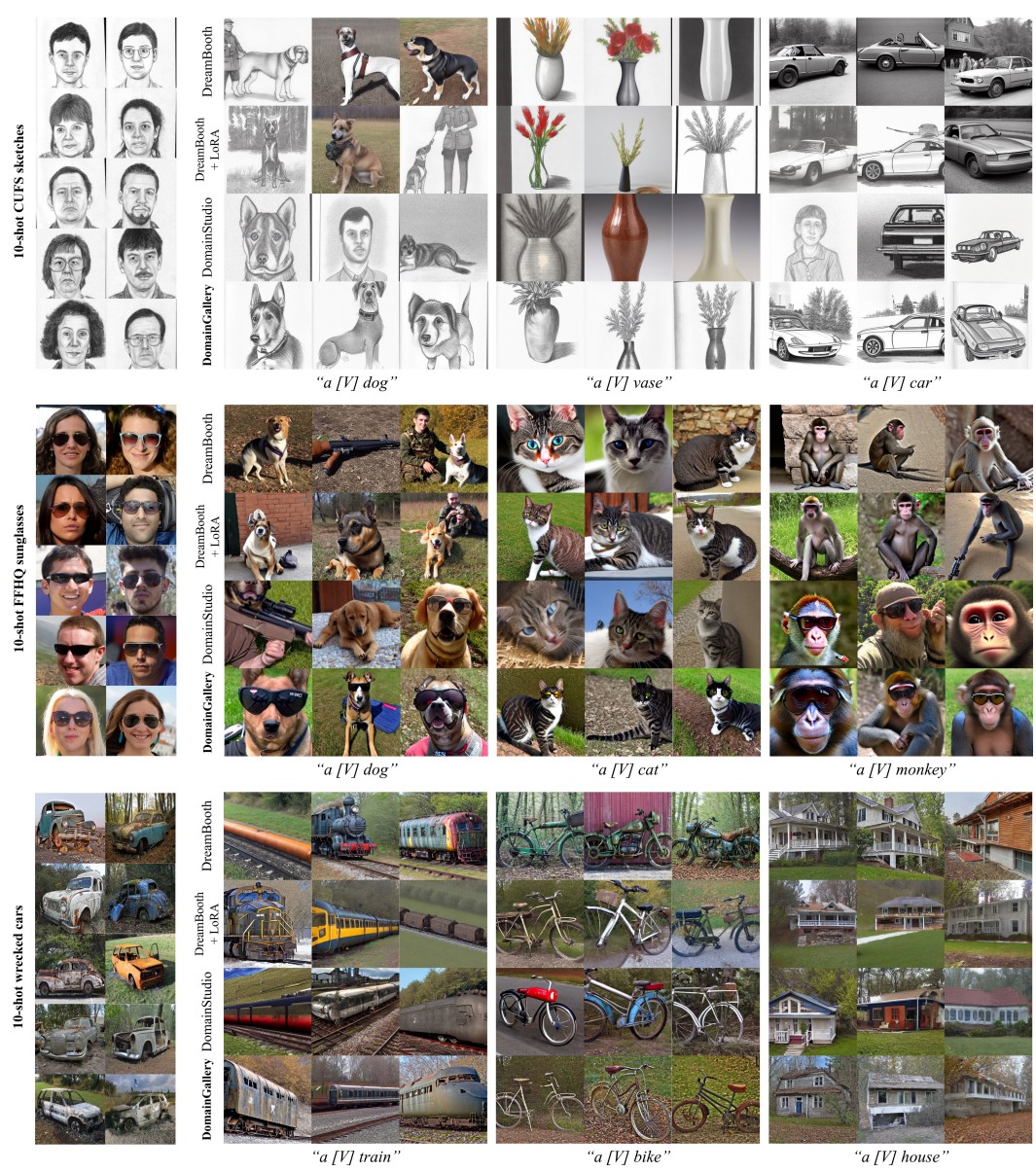

Figure 11: The 10-shot datasets (left) and the cross-category samples generated by the baselines and DomainGallery (right), on CUFS sketches (top), FFHQ sunglasses (middle) and wrecked cars (bottom).

