# OpenReview forum: "DomainGallery: Few-shot Domain-driven Image Generation by Attribute-centric Finetuning"
_NeurIPS.cc/2024/Conference — NeurIPS 2024 poster_

### Official Review · Reviewer_N3qa · 2024-07-10

**Soundness:** 3
**Presentation:** 4
**Contribution:** 3
**Rating:** 6
**Confidence:** 5

**Summary:**

This paper tackles the problem of few-shot image generation, in a scenario where it is domain-driven and defined by several attributes shared among different target data.

**Strengths:**

1. The research topic is of general interest in the generative models, and this is a good bridge between subject-driven generation and the domain-driven generation, using few-shot training data.

2. The proposed method/framework, DomainGallery, is shown effective on different generation scenario.

3. The experiment results are somehow sufficient, good-looking, both quantitatively (FID,KID, i-LPIPS) and qualitatively (visualization in different setup).

**Weaknesses:**

1. The base models covered in this work are limited. In fact, while Dreambooth is one of the exemplar baseline for few-shot text-to-image (T2I) models, there are more latest and advanced T2I models also suitable for this study. I would suggest the authors to cover 1-2 more advanced T2I models to demonstrate that the proposed DomainGallery framework is a generalized method and applicable to different model structure / design.

2. I did not see any discussion / visualization of the possible failure cases. It is better to include the failure examples to show the boundary of the proposed method.

**Questions:**

1. I would like to ask if the authors consider the mixture of style/attributes, in contrast to the multi-subject customization work, using different unique identifiers?
Indeed, I do see the mixture of [style] and [subject] results in Figure 6, but it is better if you can include the results of mixture-of-style as the supplement.

2. Are visualization results in the experiments obtained via fixed/deterministic sampling and fixed input/noise? In this way, we can compare different methods clearly and fairly.

**Limitations:**

The authors discussed the limitations and broader impact properly in the appendix C and D.

Meanwhile, I appreciate that the authors acknowledge the cherry-picking and issues of generated images on the Sunglasses dataset, as mentioned in the limitation section.

---

> ### Author Rebuttal · Authors · 2024-08-06
>
> ### Weakness 1
>
> We are not quite sure whether the phrase *more latest and advanced T2I models* mentioned by the reviewer refers to the newer subject-driven methods after DreamBooth, or newer T2I models such as Stable Diffusion after version 1.4 we used in our work.
> - **If it refers to newer subject-driven methods**, we would like to thank the reviewer for this suggestion, as applying the designs of DomainGallery on newer subject-driven methods is indeed an interesting topic and may bring some extra benefits. However, we are afraid that it is slightly beyond the scope of our work, which focuses on domain-driven generation rather than subject-driven generation. To the best of our knowledge, none of the recent subject-driven methods has recognized and tried to settle the main challenges that our work has discussed and solved, such as prior attributes, attribute entanglement etc. Therefore, we suppose that using newer subject-driven methods as our base models is unlikely to significantly boost the performance on domain-driven generation scenarios. Nevertheless, we are glad to give it a try in future work, as we have mentioned in Line 232 of our paper, *while we suppose that there may be a more delicate way to equip DomainGallery with subject-driven methods, we would like to leave it for future works*.
> - **If it refers to newer T2I models**, as we have mentioned in Line 432, *DomainGallery is probably applicable to newer versions since the attribute-centric techniques proposed in this work are not based on specific structures*. If a newer T2I model (1) can be finetuned by setting identifier words similar to DreamBooth; (2) can be equipped with LoRA; and (3) follows the conventional single-timestep training procedure of ordinary diffusion models, then the idea of DomainGallery is very likely to be applicable to it. Since it takes time to adapt our DomainGallery to other base models, it is regretful that we are not able to present some results during this response and discussion period. However, we are still grateful for this suggestion and we will give it a try in future work.
>
>
> ### Weakness 2
>
> Actually we have already discussed some possible failure cases, including learning on composite domains (rather than single domain with clear common attributes) and learning on content domains (see Appendix C). It is true that less satisfying samples may also show up sometimes. In the PDF file in the global response (Fig. 3), we have illustrated some uncurated images. Currently, our DomainGallery is indeed not yet a 100% perfect model (just as any other generative model). However, we have already made great progress in settling the challenges mentioned in our paper, such as prior attributes, attribute entanglement, and inadequate strengths of attributes.
>
>
> ### Question 1
>
> In the PDF file in the global response (Fig. 2), we have presented some generated images with mixed styles of CUFS sketches and Van Gogh houses, where we load both LoRA modules separately learned on the two datasets and use the shared identifier [V] via *"a [V] face"*. These results have shown that we may generate some images with fused attributes from both domains. However, as also mentioned by Reviewer 2oCE, there are likely conflicting attributes between two domains of styles (e.g. the sketches are colorless while the painting style of Van Gogh has strong color). Hence, mixing multiple styles is sometimes not a well-defined problem. Although it may make some interesting results (such as what we show in Fig. 2), we still suggest handling it with care.
>
>
> ### Question 2
>
> The samples in our work are generated via deterministic DDIM sampler, yet possibly from different input noises. Although it may not be the fairest way for a qualitative comparison, due to the page limitation, we are only able to present just a couple of samples for each setting. Therefore, we may have slightly picked some representative samples that manifest the typical quality of the images in order to show more clearly the drawbacks of the previous methods and how our method solves them.
>
> In the PDF file in the global response (Fig. 3), we have shown some uncurated qualitative results. From the results we may conclude that, though not yet perfect, our DomainGallery has already made great progress that most of the generated images are better than those produced by previous works w.r.t. both styles and contents.
>
>
> We will try our best to settle the reviewers' concerns. Please feel free to leave further comments. Thanks!

---

> > ### Comment · Reviewer_N3qa · 2024-08-13
> > **Acknowledgement**
> >
> > I acknowledge and have read the author response.
> >
> > In the next version, I would like to see if the authors can improve the presentation and make it easier to understand the motivation and illustration of the proposed algorithm, esp the attribute erasure part.
> >
> > Overall, I thank for the author response and would like to keep my score.

---

> > > ### Author Response · Authors · 2024-08-13
> > >
> > > Thank you very much for your approval! We will accordingly complement our work with what you suggest in the next version.
> > >
> > > If there is anywhere we can make further improvement or provide additional information, that might possibly let you consider a higher rating, please let us know.

---

### Official Review · Reviewer_btQa · 2024-07-11

**Soundness:** 2
**Presentation:** 3
**Contribution:** 2
**Rating:** 5
**Confidence:** 4

**Summary:**

This paper introduces a few-shot domain-driven image generation method that finetunes pretrained Stable Diffusion in an attribute-centric manner. It features prior attribute erasure, attribute disentanglement, attribute regularization and attribute enhancement.

**Strengths:**

1. The results seem good without overfitting.
2. Overall the paper is well-written and easy to follow.

**Weaknesses:**

1. The similarity consistency loss has been utilized in GAN-based methods. It is similar to “Few-shot image generation via cross-domain correspondence”[1].
2. The comparative experiments for personalization are missing, such as textual inversion.
3. The concept of multi-attribute transfer is quite similar to hybrid domain adaptation[2]. It needs to include a conceptual comparison in the related work section.

> [1] Ojha U, Li Y, Lu J, et al. Few-shot image generation via cross-domain correspondence[C]//Proceedings of the IEEE/CVF conference on computer vision and pattern recognition. 2021: 10743-10752.

> [2] Li H, Liu Y, Xia L, et al. Few-shot Hybrid Domain Adaptation of Image Generator[C]//The Twelfth International Conference on Learning Representations. 2023.

**Questions:**

As seen in weakness.

**Limitations:**

The manuscript includes discussions on limitations and social impacts.

---

> ### Author Rebuttal · Authors · 2024-08-06
>
> ### Weakness 1
>
> With regard to the part of attribute regularization, we have utlized a variant of the similarity consistency loss in CDC [1]. However, our core contribution of this part is not the similarity consistency loss itself, but is the strategy to construct **paired** source/target codes in **latent space** via **few-step recurrent denoising** within the single-timestep training procedure of diffusion models, so that our DomainGallery is open to various choices of regularizations (including but not limited to the similarity consistency loss we apply in our model). Without such strategy, the previous work DomainStudio directly applied regularization on **single-step-denoised, unpaired, and pixel-level** source/target images, which we have discussed and proved to be much less effective. Refer to Line 189--194.
>
>
> ### Weakness 2
>
> In the generation scenario of personalization, the reason we combine DomainGallery and DreamBooth is just to show that it is promising to combine DomainGallery and subject-driven methods to achieve personalization of certain domain and certain subject. In this work, our goal is still to focus on domain-driven image generation and accordingly design DomainGallery, rather than trying to combine DomainGallery and a variety of existing subject-driven methods (including Textual Inversion [2]) to see which performs the best, which is slightly beyond the scope of this work. However, we still would like to thank the reviewer for raising this suggestion. Just as we have mentioned in Line 232, *while we suppose that there may be a more delicate way to equip DomainGallery with subject-driven methods, we would like to leave it for future works*.
>
>
> ### Weakness 3
>
> Thanks for recommending Hybrid Domain Adaptation (HDA), which is an insightful work. However, after reading it, we are afraid that the connection between HDA and our DomainGallery is not that strong. Here we list some key differences between them.
>
> - **Setting:** HDA focuses on transfering a model to a hybrid domain defined by multiple independent domains, while our work mainly focus on transferring to a certain **single** domain, which is the mainstream setting for domain transfer. (Also we humbly discourage the use of the term *multi-attribute transfer*, since a single domain usually contains multiple attributes as well.) It is also noteworthy that we have already mentioned and discussed many previous works on (single) domain transfer in Sec. 1 and Sec. 2 (e.g. Line 25--30, 64--70).
> - **Model:** Our DomainGallery is based on a pretrained Text-to-image diffusion model (i.e. Stable Diffusion), while HDA is a GAN-based work. Also, the designs of HDA cannot be directly applied to diffusion models either, especially the latent diffusion models adopting single-timestep training procedure, since iteratively denoising latent codes and decoding them into pixel-level images usually brings unacceptable time and space consumption during training periods.
> - **Generation scenario:** Since HDA adopts GAN, which follows a noise-to-image pipeline, it is only capable of generating intra-category images without further control, like the first scenario of the four in our work. On the contrary, our DomainGallery can take the benefits of the text control, and master in the other three generation scenarios (cross-category, extra attributes, personalization).
>
>
> [1] U. Ojha et al., Few-shot Image Generation via Cross-domain Correspondence.
>
> [2] R. Gal et al., An Image is Worth One Word: Personalizing Text-to-Image Generation using Textual Inversion.
>
>
> We will try our best to settle the reviewers' concerns. Please feel free to leave further comments. Thanks!

---

> > ### Comment · Reviewer_btQa · 2024-08-13
> >
> > Thank you for your detailed rebuttal. I appreciate the effort you put into addressing all of my concerns. As a result, I have decided to increase my score.
> >
> > I would also like to acknowledge the authors for discussing the difference with previous works and conducting additional experiments. I believe these discussions offer valuable insights and would enhance the article. Therefore, I strongly suggest including them in the camera-ready version of the paper.

---

> > > ### Author Response · Authors · 2024-08-13
> > >
> > > Glad to hear that all of your concerns are resolved, and thank you very much for reconsidering your rating.
> > >
> > > We will follow your advice to include what we have discussed in the next version.

---

### Official Review · Reviewer_Ue4P · 2024-07-12

**Soundness:** 3
**Presentation:** 3
**Contribution:** 2
**Rating:** 5
**Confidence:** 3

**Summary:**

Text-to-image generation models pre-trained on large-scale datasets made progress but the models are still limited when we expect to generate images that fall into specific domain or style that are hard to describe or unseen to models. This paper proposed the DomainGallery, a few-shot domain-driven image generation method which aims at finetuning pre-trained stable diffusion on few-shot target datasets in an attribute-centric manner. The proposed method focuses on attribute operations, including erasure, disentanglement, regularization and enhancement.

**Strengths:**

-	The paper is well-structured and well-motivated.
-	The method seems sound and reasonable.
-	The problem to tackle is a sensible and essential problem.

**Weaknesses:**

-	It seems to me that there are several pipelines and tasks to perform in this paper, but the technical challenges should be illustrated in a concise and precise manner. It seems to me that the engineering solutions and objectives are clearly illustrates but the technical challenges are not summarized well. In this way, the technical contributions seem weak and unclear.
-	The proposed method seems like a combination of existing techniques. The novelty seems limited unless the authors argue specifically.
-	There seems to lack specific error analysis on whether the result improvement is due to the proposed method. It would be very important to demonstrate this, either quantitatively, or qualitatively. At least, in different dataset directions, e.g skeches, sunglasses. shall we demonstrate what exactly causes the improvements? Why improvements on which samples in the dataset?

**Questions:**

Please reply with points in the weakness.

**Limitations:**

Please see weakness

---

> ### Author Rebuttal · Authors · 2024-08-06
>
> ### Weakness 1
>
> Thanks for the advice. Here we recap the four technical challenges we have mainly focused on in our work:
>
> - **Challenge 1:** Prior attributes of the identifier [V] may show up in the generated samples even if we have bound new domain attributes to it.
> - **Challenge 2:** The identifier [V] and the category word [N] may sometimes leaks some attributes into each other.
> - **Challenge 3:** The model is prone to overfitting when finetuned on few-shot datasets.
> - **Challenge 4:** Sometimes the strengths of the learned domain attributes are inadequate for cross-category generation scenarios.
>
> In order to let the readers better comprehend these technical challenges and how DomainGallery settles them, in Sec. 4, we chose to discuss these four challenges and correspondingly our four solutions **one by one** from Sec 4.1 to 4.4 (also see the four techniques in our response to Weakness 2 below). Due to page limitation, we were not able to give detailed description in the introduction, yet we briefly mentioned them in Line 52--57 when we pointed out our four attribute-centric techniques. We will follow the reviewer's advice and deliver a better and clearer summary on these challenges in the next version.
>
>
> ### Weakness 2
>
> We would like to further clarify the novelty of our DomainGallery and its four attribute-centric finetuning techniques as below, which respectively settle the four challenges summarized in our response to Weakness 1 above.
>
> - **Technique 1: Prior attribute erasure & Technique 2: attribute disentanglement:** To the best of our knowledge, no previous work (either on domain-driven or subject-driven generation) has adopted similar strategies. As explained in Line 141 and 171, these two issues do not even exist in subject-driven generation, since the identifier [V] and the category word [N] will always be paired together. While in domain-driven generation where they **are** problems indeed, the only previous method (i.e. DomainStudio) failed to recognize and try to solve these two issues either.
> - **Technique 3: Attribute regularziation:** With regard to this part, our core contribution is not the similarity consistency loss itself, but is the strategy to construct **paired** source/target codes in **latent space** via **few-step recurrent denoising** within the single-timestep training procedure of diffusion models, so that our DomainGallery is open to various choices of regularizations (including but not limited to the similarity consistency loss we apply in our model). Without such strategy, the previous work DomainStudio directly applied regularization on **single-step-denoised, unpaired, and pixel-level** source/target images, which we have discussed and proved to be much less effective. Refer to Line 189--194.
> - **Technique 4: Attribute enhancement:** There are indeed some previous works (though not in domain-driven or subject-driven generation) adopting similar methods to enhance certain attributes. However, in this part we have proposed two modes (VN-N & V-uncond) of attribute enhancement, and have verified the effectiveness of our final choice (i.e. V-uncond) throughout the ablation studies in Appendix B.1.
>
>
> ### Weakness 3
>
> About whether the result improvement is due to the attribute-centric finetuning techniques we propose, we have already presented **extensive ablation studies in Appendix B.1**, proving the necessity and the effectiveness of each part of our design with both qualitative and quantitative results (also approved by Reviewer 2oCE in Strength 3). Due to page limitation, we were not able to put them in the main paper. However, we do recommend reading this part to get a better understanding of our designs.
>
>
> We will try our best to settle the reviewers' concerns. Please feel free to leave further comments. Thanks!

---

### Official Review · Reviewer_2oCE · 2024-07-12

**Soundness:** 3
**Presentation:** 3
**Contribution:** 3
**Rating:** 6
**Confidence:** 4

**Summary:**

This paper addresses the few-shot domain transfer problem in text-to-image diffusion generation, where the goal is to keep the style and attributes of the source examples while being able to generate images of potentially different subject matters. It binds the attributes to rarely-used tokens, but observes that these tokens could potentially induce correlated unintended features, and it adopts a prior-attribute erasure procedure to remove such correlation. It then finetunes the model for domain transfer, enforcing disentanglement of attributes and subject categories and applying attribute regularization. Additionally, for cross-category generation, it proposes an attribute enhancement technique to make the learned attributes more prominent in the generations.

**Strengths:**

* The paper investigates an interesting task of attribute-centric transfer.
* The method is well-motivated and conceptually solid. The paper observes limitations in existing approaches, such as the attribute correlation with binding tokens, and entanglement between attributes and subject categories, and effectively addresses these limitations in the proposed method.
* The experiments consider a variety of settings such as intra- and inter-category transfer, and transfer with additional attribute or specific subject identity. They encompass a variety of domains and settings, and provide numerous qualitative visualizations to facilitate understanding of the method performance. The ablation studies effectively verify the contribution of each component.

**Weaknesses:**

* The paper has discussed most of the limitations of the method, such as in cases where the examples contain multiple subject categories or multiple sub-domains, or in cases with certain local/content attributes.

* One additional case is that as the method infers the attributes infers from the examples, sometimes it can pick up unintended attributes that the few-shot examples happen to share (e.g. for FFHQ glasses, cross-category generations mostly have outdoor/nature backgrounds, even though this may not be necessarily intended for this setting). It would be interesting to see extensions that allow more explicit control over including/excluding certain attributes.

**Questions:**

* It would be helpful to have more clarifications on whether the qualitative results mostly randomly selected or hand-picked, especially given that the performance of cross-category generations are primarily demonstrated through qualitative visualizations. The paper mentions that the cross-category generations for FFHQ sunglasses in Fig 11 involved some cherry-picking, but does not seem to specify for the others.

* For better understanding of the proposed method's behavior, it would interesting to explore the boundary for what attributes get encoded into the binding tokens. For example, when all examples contain one subject centered in the pictures, whether/to what extent the number, position or subject size are considered part of the attributes and get picked up by [V]. It would also be interesting to see how the method behaves when the textual prompt contains conflicting attributes with the learned binding token (e.g. "A [V] bird with blue feathers" where [V] is learned from colorless sketches).

**Limitations:**

The authors have adequately discussed the limitations and societal impact.

---

> ### Author Rebuttal · Authors · 2024-08-06
>
> ### Weakness 1 & 2
>
> We would like to thank the reviewer for raising concerns about the *unintended attributes*, which is indeed one of the core issues in few-shot image generation. Due to the lack of sufficient data samples, there is usually a gap, either large or little, between the distribution of the (few-shot) dataset and the (agnostic) distribution of the domain. For example, an unexpected attribute that happens to be shared among all the dataset images, as mentioned by the reviewer. (With regard to this topic, we recommend a survey on few-shot image generation. [1])
>
> As in Line 102 of the paper, we define a (dataset-derived) domain as *the common attributes shared by all the images*. Therefore, all the attributes will be encoded into the identifier as long as they are shared among the images, though some of them may be unexpected (e.g. outdoor backgrounds shared by the sunglasses dataset). From this point of view, our DomainGallery still complies with such definition.
>
> If we intend to manually exclude some unwanted attributes, a possible solution is to put them into the text prompt when finetuning the model, so that they will not be encoded into the domain identifier [V], just as we explicitly exclude the shared categorical attribute by adding a category word [N]. Since it takes time to adapt our DomainGallery to such setting, it is regretful that we are not able to present some results during this response and discussion period. However, we are still grateful for this suggestion and we will do research on this topic in future work.
>
>
> ### Question 1
>
> Except for the case of sunglasses described in the limitations in Appendix C, our qualitative results are indeed mostly randomly selected. Due to the page limitation, we are only able to present just a couple of samples for each setting, so we may have slightly picked some representative samples that manifest the typical quality of the images in order to show more clearly the drawbacks of the previous methods and how our method solves them.
>
> In the PDF file in the global response (Fig. 3), we have shown some **uncurated** qualitative results. Although our DomainGallery is not yet a 100% perfect model (just as any other generative model), we have already made great progress in settling the challenges mentioned in our paper, such as prior attributes, attribute entanglement, and inadequate strengths of attributes.
>
>
> ### Question 2
>
> Just as our response to Weakness 1 & 2 says, our model will learn all the common attributes into [V]. Thus if all the images of a certain dataset contain identical or similar number, position, or size of the subject, they will be included in [V] as well. As in Fig. 3 of our paper shows, all the generated samples have one face in the center, with a medium size and a frontal or nearly-frontal view.
>
> About possible conflicting attributes in [V] and the text prompt, we have illustrated some samples in the PDF file in the global response (Fig. 1) with prompt *"a [V] bird with blue feathers"* where [V] is learned from CUFS sketches, as the reviewer suggested. Although there are conflicting attributes, our DomainGallery seems to be capable of generating images with partially fused attributes. While the images are generally in grayscale, some blue feathers still appear. Although such trials may bring some interesting outcomes, we suggest handling conflicting attributes with care.
>
>
> [1] Z. Li et al., A Comprehensive Survey on Data-Efficient GANs in Image Generation.
>
> We will try our best to settle the reviewers' concerns. Please feel free to leave further comments. Thanks!

---

> > ### Comment · Reviewer_2oCE · 2024-08-12
> >
> > Thank the authors for the rebuttal. I am satisfied with the response and decide to keep my score.

---

> > > ### Author Response · Authors · 2024-08-12
> > >
> > > Thanks for your reply, and we sincerely appreciate your support. If there is anywhere we can make further improvement or provide additional information, that might possibly let you consider a higher rating, please let us know. Thank you very much!

---

### Author Rebuttal · Authors · 2024-08-06

### Recap

In this work, we propose DomainGallery, a method for few-shot domain-driven image generation, which analyzes the key issues that previous works failed to settle, and accordingly designs a series of attribute-centric finetuning techniques. With these novel and effective techniques, our DomainGallery greatly outperforms SOTA works in several generation scenarios.

### Global Response to All Reviewers

First we would like to express our sincere gratitude to all the reviewers for spending time and effort reviewing our paper, coming up with insightful comments and suggestions, and helping us make our work better. We especially appreciate that several strengths of our work have been recognized by the reviewers, listed as below.

- Our method is well-motivated, conceptually solid, sound, and reasonable. (Reviewer 2oCE, Ue4P)
- The problem and the task we focus on are sensible, essential, and of general interest in the generative models. (Reviewer 2oCE, Ue4P, N3qa)
- Our work effectively addresses the limitation of existing methods, thus achieving good performance. (Reviewer 2oCE, btQa, N3qa)
- The evaluation of our work encompasses a variety of settings, with numerous experimental results, and has sufficient ablation studies that verify the contribution. (Reviewer 2oCE, N3qa)
- The paper is well-written, well-structed and easy to follow. (Reviewer Ue4P, btQa)

Later on, we will reply to the reviewers' concerns separately. **Please also check on the PDF file below which contains some figures as a part of our response.**

---

### Author Response · Authors · 2024-08-14
**To ACs and Reviewers: A Summary of the Discussions**

At the end of the discussion period, we would like to thank you all for your time and effort. We are delighted that our DomainGallery has received many insightful comments, as well as common approval from all the reviewers. We believe that these discussions will definitely make our work a better one.

We have already summarized the strengths of our work that recognized by the reviewers in the *Global Response to All Reviewers*. For your convenience, here we further collate the main concerns raised by the reviewers and our responses to them, and present them as below.

### Reviewer 2oCE

- *Possible unexpected common attributes*: We have clarified that learning all the common attributes (either expected or unexpected) complies with the definition of domain-driven generation. We have also provided a possible solution to explicitly exclude unwanted attributes similar to what we have done to categorical attributes.
- *Qualitative result selection*: We have illustrated some uncurated samples in the rebuttal PDF, demonstrating that the images generated by DomainGallery generally have good quality.
- *Possible conflicting attributes*: We have shown some examples in the rebuttal PDF, where our DomainGallery has still produced some interesting results with mixed attributes.

### Reviewer Ue4P

- *Technical challenges*: We have provided a summary of the four technical challenges in our work, and we will accordingly further clarify this part in the next version.
- *Contribution and novelty*: We have explained that no previous work recognized and tried to settle these technical challenges, therefore our DomainGallery is indeed novel and may contribute to the whole community of generative models.
- *Ablation study*: We have clarified that actually we have already conducted extensive ablation studies in Appendix B.1.

### Reviewer btQa

- *Novelty of the similarity consistency loss*: We have explained that our contribution w.r.t. this part is actually the proposed strategy constructing paired source/target latent codes, so that our method is open to a variety of regularizations not limited to the similarity consistency loss we have adopted in this work.
- *Comparison with subject-driven methods*: We have clarified that our goal is to mainly focus on domain-driven generation, while combining DomainGallery with personalization methods is just one of the promising directions and we prefer to leave this part to future works.
- *Connection to Hybrid Domain Adaptation (HDA)*: We have discussed some key differences (incl. settings, models and generation scenarios) between HDA and our DomainGallery.

### Reviewer N3qa

- *Applying DomainGallery to other subject-driven methods/T2I models*: These two are indeed promising future directions. We have respectively discussed our opinions (with some predictions on performance) towards these two kinds of potential combinations.
- *Possible limitations & qualitative comparison*: We have clarified that we have actually discussed possible limitations in Appendix C. And we have also illustrated uncurated samples in the rebuttal PDF, including some less satisfying samples. Although not yet perfect, our DomainGallery generally performs well.
- *Mixed domains*: We have shown some samples mixing two style domains in the rebuttal PDF, demonstrating that DomainGallery may produce some interesting results.

We will accordingly improve our manuscript following the suggestions from the reviewers. Again we would like to express our gratitude to their precious comments!

---

### Decision · Program_Chairs · 2024-09-25

**Decision:**

Accept (poster)

**Comment:**

This paper proposes a few-shot domain-driven image generation method which tries to finetune the pretrained Stable Diffusion on few-shot target datasets in an attribute-centric manner. It supports features including prior attribute erasure, attribute disentanglement, attribute regularization and attribute enhancement. The reviewers like this paper due to 1) the studied task is interesting; 2) the method is well-motivated and solid; 3) both qualitative and quantitative results look promising; and 4) the writing is clear and easy to follow. There are some minor concerns on the novelty, missing additional explanations and analysis, etc. But the rebuttal has resolved most of them, and the reviewers are satisfied in the end. The authors should include the reviewers' suggestions in the final version. (If the authors think some additional suggested references are not very related to this paper, feel free to ignore them.)